# A Capsaicin-Based Phytogenic Solution Improves Performance and Thermal Tolerance of Heat-Stressed Growing Pigs

**DOI:** 10.3390/ani14060973

**Published:** 2024-03-21

**Authors:** Miguel Cervantes, Panagiotis Sakkas, Moisés Soto, Alejandra Jaquelin Gómez, Reyna L. Camacho, Néstor Arce, Nicolas Quilichini, Adriana Morales

**Affiliations:** 1Instituto de Ciencias Agrícolas, Universidad Autónoma de Baja California, Mexicali 21100, Mexico; miguel_cervantes@uabc.edu.mx (M.C.); moises.soto43@uabc.edu.mx (M.S.); nestor.arce@uabc.edu.mx (N.A.); 2Laboratory of Nutrition, Faculty of Veterinary Medicine, Aristotle University, 54124 Thessaloniki, Greece; psakk@vet.auth.gr; 3DELTAVIT, CCPA Group, Z.A. du Bois de Teillay, 35150 Janzé, France; nquilichini@ccpa.com

**Keywords:** heat stress, growing pigs, performance, thermoregulation, oxidative stress, AA metabolism, *Capsicum* spp., phytogenics

## Abstract

**Simple Summary:**

Exposure to heat stress significantly impairs the productive performance of growing pigs. Supplementing their diet with phytogenic solutions, especially those containing *Capsicum* spp., can boost their thermal tolerance due to the activities of its principal active metabolite, capsaicin. This improvement aids in better nutrient consumption and efficient utilization for growth. In this study, we explored the effectiveness of a *Capsicum* spp.-based dietary phytogenic solution in alleviating the effects of heat stress on pigs. Our findings showed that this supplementation notably enhanced the pigs’ performance under heat stress conditions by increasing feed intake and improving feed efficiency. It also positively affected thermoregulatory responses, as evidenced by lowered body temperatures. Moreover, the supplement improved antioxidant defenses and heat shock protein responses and bolstered intestinal integrity and post-absorptive metabolism. Overall, the results indicate that a dietary phytogenic solution derived from *Capsicum* spp. can effectively counteract some of the detrimental effects of heat stress in pigs, leading to improved overall productive performance.

**Abstract:**

Exposure to heat stress (HS) detrimentally affects pig performance. This study explored whether a dietary phytogenic solution based on *Capsicum* spp. (PHY) could enhance the thermal tolerance of heat-stressed growing pigs. Forty-two individually housed pigs were randomly assigned to three treatments: thermoneutral pigs on a control diet (TN-C) and pigs subjected to HS fed the control diet either without (HS-C) or with supplemental PHY (HS-PHY). The TN-C group exhibited increased average daily gain (ADG) and feed intake (FI) compared to both HS-C (*p* < 0.01) and HS-PHY pigs (*p* < 0.05) and better feed efficiency compared to HS-C pigs only (*p* < 0.01). However, the HS-PHY pigs showed significantly higher FI (*p* < 0.01) and ADG (*p* < 0.05) compared to HS-C pigs. HS pigs displayed higher body temperatures (BTs) than TN pigs (*p* < 0.01), yet HS-PHY pigs experienced a lesser increase in BT compared to HS-C pigs (*p* < 0.05). Supplementation with PHY mitigated some effects of HS, increasing serum superoxide dismutase and catalase activity, reducing HSP90 expression in *longissimus dorsi* muscle, and elevating jejunal villus height compared to HS-C pigs (*p* < 0.05), reaching levels akin to TN-C pigs. Additionally, PHY supplementation resulted in lower serum urea levels than HS-C pigs (*p* < 0.01) and similar myosin gene expression to TN-C pigs (*p* > 0.1), suggesting enhanced amino acid post-absorptive utilization for lean tissue growth. In conclusion, dietary PHY supplementation partially offset the adverse effects of HS on pig performance by improving thermal tolerance.

## 1. Introduction

When climatic conditions are above the animal’s thermal comfort zone (upper critical temperature threshold), animals experience heat stress (HS) as they fail to dissipate heat efficiently to the environment, leading to the initiation of physiological and metabolic adaptation mechanisms. These adaptations invariably compromise their productive potential [1]. Significant economic losses are associated with HS, as the majority of pork production occurs in regions where pigs are housed either temporarily (during summer) or permanently under HS conditions [2]

Pigs exposed to HS may experience an increase of up to 2.5 °C in their body temperature (BT) [3]. A highly conserved animal response to HS present in livestock species, including pigs, is the induction of anorexia (i.e., a voluntary reduction of feed intake) to decrease the internal heat production deriving from nutrient digestion and metabolism, which in turn adversely affects their productivity [4]. Concurrently, pigs experiencing HS redirect blood flow to the periphery to enhance body heat dissipation [5]. However, this flow redirection reduces the amount of blood reaching internal organs [6], leading to reduced oxygen and nutrient delivery to the small intestine and resulting in epithelial damage (shortened intestinal villus height; [3]). The reduced intestinal villus height in HS pigs has been linked to increased production of reactive oxygen species (ROS) and oxidative stress [2], leading to mitochondrial damage [7] and subsequent intestinal cell death further damaging the epithelia and compromising their integrity [8]. It has been reported that HS-related oxidative stress increment negatively affects the digestive–absorptive function of the small intestine, as evidenced by increased losses of endogenous amino acids (AAs), reduced AA digestibility [9], depressed expression of genes coding for the synthesis of AA transporters [10], and lower concentrations of free AAs in serum. Moreover, the permeability of the intestinal epithelia appears to be compromised [11], which eventually might affect the health of HS pigs. Thus, increasing the activity of the cellular antioxidant system is expected to counteract the negative impact of exposure to HS.

Phytogenic feed additives containing plant secondary metabolites (PSMs) may mitigate the adverse effects of heat stress. The number of studies investigating the effects of PSMs on HS tolerance in growing–finishing pigs is rather limited [12] compared to other livestock species, including poultry, with variable effects reported [13,14]. Of particular interest regarding their capacity to affect thermoregulatory responses in pigs and ruminants are capsaicinoids, especially capsaicin, which are alkaloids that are found in chili peppers (*Capsicum* spp.) and are responsible for their burning and irritant effects [15]. Capsaicin is a prototypical transient receptor potential cation channel subfamily V member 1 (TRVP-1) agonist, which serves as one of the principal thermosensors for the thermoregulatory system, being conserved across mammalian species [16]. Activation of TRVP-1 by capsaicin lowers body temperature and induces hypothermia via peripheral vasodilation [17], potentially increasing the animal’s upper critical temperature threshold and therefore its thermal tolerance [18]. Studies in growing–finishing pigs have shown that capsaicin-containing additives may improve feed efficiency following both short- and long-term HS exposure, although effects on thermoregulatory responses, such as respiration rate (RR), were limited under short-term HS exposure [19,20]. Capsaicin-containing additives have also been demonstrated to be effective in improving the productivity of heat-stressed dairy cows [21,22,23] although effects on feed intake (FI) and rectal temperatures have not been consistent [21,22]. Further to its actions on thermoregulation, capsaicin may ameliorate oxidative stress and associated inflammatory responses [24], while it has pronounced effects on nutrient digestion, independent of HS exposure [25,26]. These effects are theoretically significant in countering the development of leaky gut [27] and could potentially alter both AA absorption and utilization, as they are influenced by both the inflamed and HS state in growing pigs [10,28], and consequently their performance.

To the best of the authors’ knowledge, this study is the first to concurrently examine the effects of a phytogenic solution derived from *Capsicum* spp. (PHY) on various aspects of pig health under heat stress (HS). Specifically, it investigates PHY’s impact on pig performance, body temperature, respiratory frequency, the activity of antioxidant enzymes, and the concentration of amino acid metabolites in blood serum. Additionally, it assesses intestinal histomorphology and the expression of genes responsible for synthesizing heat shock protein 90 (HSP90) in the liver and muscle and myosin in muscle. We hypothesized that dietary supplementation with PHY can mitigate some of the adverse effects of HS in pigs. The anticipated benefits include modulating thermoregulatory responses to lower body temperature, reducing oxidative stress, and improving intestinal integrity. Consequently, we expect that the performance of pigs subjected to HS will be enhanced with PHY supplementation.

## 2. Materials and Methods

### 2.1. General

The pigs used in the present study were cared for in accordance with the guidelines established in the Official Mexican Regulations on Animal Care [29] and approved by the Ethical Committee of the Institute of Agricultural Sciences at Universidad Autónoma de Baja California. The experiment was conducted in the Metabolism and Physiology Unit of our university, located in Mexicali, Baja California, in northwest Mexico (geographical location 32°24′ N and 115°11′ W), during July and August, the warmest season of the year. All pigs were individually housed in 1.2 × 1.2 m pens inside a temperature-controlled room or a room without temperature control with the typical ambient temperature (AT) fluctuations that provoke HS. Each pen had a raised slated plastic floor and was equipped with a single-hole feeder and a water nipple to allow ad libitum consumption of feed and water. The HS room was naturally ventilated by opening the windows, and the AT inside the TN room was controlled with the aid of an A/C unit with the thermostat set at 22 ± 2 °C; the light was turned on all the time. According to the Federation of Animal Science Societies, the TN zone for pigs within the 25 to 50 kg range is around 22 °C [30]. The AT and relative humidity were measured with the aid of small devices (Hygrothermographs; Thermotracker Inc. iButtonLink LLC, Whitewater, WI, USA) installed inside each of the rooms and set to record these variables every hour. The temperature–humidity index was calculated according to Rothfusz [31].

### 2.2. Animals, Diet, and Experimental Procedure

Forty-two crossbred (Landrace × Hampshire × Duroc) pigs with an average initial BW of around 27.0 kg were distributed in three groups based on initial body weight, sex, and litter, randomly assigned to three treatment groups; there were 14 replicates per treatment. One group of pigs was housed under thermoneutral (TN) conditions inside the temperature-controlled room and fed a typical wheat–soybean meal control diet (TN-C). The other two groups were housed under HS conditions inside the room with no ambient temperature control. One of these groups was fed the same control diet as the TN group (HS-C), but the other group of pigs was fed the control diet supplemented with the additive (HS-PHY). All pigs had free access to feed and purified water all the time. In addition, thermographs (Thermotracker Inc.) were implanted into the small intestine of another group of 12 ileal-cannulated pigs, four pigs per treatment, to record BT every 15 min during the whole experiment. A thermographic camera was also used to record images and temperature from the pig’s body. All animals had a 10-day adaptation period to the pens under TN conditions, followed by an 8-day experimental period.

The control diet was formulated with wheat and soybean meal, as well as free Lys, Met, and Thr to meet the SID AA requirements [32] for pigs in the BW range of 25–50 kg (Table 1). The HS-PHY diet was the control diet supplemented with 2 g/kg of the phytogenic solution. The additive PHY consisted of a standardized plant phytogenic solution containing *Capsicum* spp. Oleoresin with capsaicin being the principal active PSM (trade name ThermoControl^®^. Janzé, France). All diets contained the same levels of SID AA [32] and were supplemented with a vitamin and mineral premix to meet or exceed the vitamin and mineral requirements for this type of pig. Fresh feed was offered in excess twice a day, at 0700 and 1900 h.

Six pigs from each treatment with body weights closest to the treatment average were subjected to fasting for 10 h, starting at 2100 h on the 8th day of the experimental period. On the 9th day of the experimental period at 0700 h, the pigs were fed 0.5 kg of their respective diet, and 2 h after the feeding, they were slaughtered by electrical stunning and exsanguination, and the corpses were eviscerated. Immediately, samples of mucosa scratched from the duodenum, jejunum, and ileum, as well as samples of liver and the *longissimus dorsi* muscle, were collected into 2 mL microtubes and rapidly stored in liquid nitrogen. The abundance of mRNA coding for the synthesis of heat shock protein 90 (HSP90) in the liver and muscle, cationic amino acid transporter 1 (CAT-1) in the liver, and myosin in muscle was analyzed. In addition, 5 cm segments of duodenum, jejunum, and ileum were collected and stored in 10% formyl buffer for later histology characterization based on the procedure described previously [33]. Blood samples were also collected during the exsanguination of the animals for determining concentrations in serum (SC) of specific AA metabolites and antioxidant capacity indicators [activity of superoxide dismutase (SOD), catalase (CAT), and glutathione peroxidase (GPX)]. Based on previous experiences, the total collection process took 10 min or less, which guarantees the good quality of the samples for RNA extraction. Blood samples were centrifuged at 1000× *g*, 4 °C, for 1 min to separate serum from blood cells. At the end of the sampling, all samples were transported to the molecular biology lab and stored at −82 °C until analysis.

### 2.3. Serum Antioxidant Activity and Amino Acid Metabolites

To measure antioxidant response, the activities of SOD, CAT, and GPX were analyzed in serum. The activities of SOD, CAT, and GPX in serum were measured by assay kits following the manufacturer’s instructions (Cayman Chemical Company, Ann Arbor, MI, USA: for SOD, assay kit # 706003; CAT, assay kit # 707002; and GPX, assay kit # 703102). Before the SC of AA metabolites was measured, serum samples were deproteinized with a Millipore Ultrafree-MC 10,000 NMWL Filter Unit (Millipore, Bedford, MA, USA) at 5000× *g*, 4 °C, for 30 min. The free radical scavenging activity of the additive PHY, the control diet with and without the additive, and the ileal digesta of the HS pigs was analyzed following the procedure by Re et al. (1999) using ABTS (2,2′-Azinobis(3-Ethylbenzothiazoline-6-Sulphonic Acid) as a control [34]. The antioxidant TPGS (D-Alpha-Tocopheryl Polyethylene Glycol Succinate) was used as a standard. The antioxidant activity expressed as the free radical scavenging activity of each sample on ABTS was calculated as the percentage of inhibition of the control by the standard. Also, the SC of AA metabolites was analyzed at the University of Missouri Chemical Labs, according to method 982.30E as described by AOAC [35]. Ion-exchange chromatography, post-column ninhydrin derivatization, and fluorescence detection at 570 nm were used for the separation and quantification of AA metabolites.

### 2.4. Gut Histomorphology

Samples of duodenum, jejunum, and ileum (5 cm each) were collected and washed with physiological saline for the determination of gut histomorphology. Samples were fixed in 10% neutral formalin for paraffin embedding. Formalin-fixed samples of the three intestinal portions were stained with hematoxylin and eosin [36]. The mucosal structure was observed using 40× magnification by HBO50 Primo Star microscopy (Zeiss, México city, México); microphotographs were obtained with a photographic camera (Canon, Tokyo, Japan) and analyzed with an Image J2 Analysis System [37]. Villus height and crypt depth of at least 5 well-oriented villi were measured per sample.

### 2.5. Gene Expression

#### 2.5.1. Total RNA Extraction and Purification

Mucosae samples from the duodenum, jejunum, and ileum were treated to extract total RNA by pulverization into liquid nitrogen following the instructions for the RNA purification kit (Direct-zol RNA Microprep Kit R2052, Zymo Research, Irvine, CA, USA) and using Trizol reagent (Invitrogen, Corp., Carlsbad, CA, USA). Purified RNA was then eluted with 30 µL of RNase-free distilled water and stored at −82 °C. The concentration of total RNA was determined spectrophotometrically (Genesys 50, Thermo Fisher Scientific Co., Rochester, NY, USA) at 260 nm, and the purity of RNA was assessed by using the A260/A280 ratio, which ranged from 1.8 to 2.0. The integrity of total RNA was evaluated by gel electrophoresis on 1% agarose gels; the RNA quality was assessed with a 28S/18S rRNA ratio around 2.0:1 [38].

#### 2.5.2. Reverse Transcription

Approximately 2 µg of total RNA were treated with 1 U of DNase I (1 U/µL; Invitrogen) and 6 µL of 5× reverse transcription buffer in a 30 µL reaction completed with nuclease-free water; the reaction was carried out for 15 min at room temperature and another 15 min at 70 °C to stop the reaction. Reverse transcription was initiated with DNase-treated RNA samples, with the addition of 1 µL of random primers (150 ng/µL, Thermo Scientific, México city, México) and 1 µL of dNTPs solution (10 µM each), 1 µL of ribonuclease inhibitor (40 U/µL; RiboLock, Thermo Scientific), 3 µL of nuclease-free water, and 2 µL of 5× reverse transcription buffer. The reaction was incubated at 42 °C for 2 min before the addition of 1 µL of reverse transcriptase enzyme (200 U/µL; RevertAid, Thermo Scientific), and then the incubation continued at 42 °C for 50 min. Finally, the mixture was incubated at 70 °C for 15 min and then chilled on ice to stop the reaction. cDNA samples were quantified spectrophotometrically and diluted to a final concentration of 50 ng/µL.

#### 2.5.3. Real-Time PCR

Specific primers for myosin heavy chain 4, cationic amino acid transporter 1 (CAT-1), and heat shock protein-90 (HSP90) mRNA were designed according to their published sequences in Genbank (Table 2). A housekeeping gene (ribosomal protein 4—RPL4) was used as an endogenous control to normalize variations in mRNA. The expression of tight junction proteins was estimated by quantitative PCR (qPCR) assays, using the Maxima SYBR Green/ROX qPCR Master Mix (Thermo Scientific) in a CFX96 Real Time System thermal cycler (BioRad, Herefordshire, UK), and quantified with the software CFX Manager 3.0 (BioRad).

### 2.6. Statistical Analysis

Analyses of variance of the data were performed based on the experimental design, using the GLM of SAS. Two contrasts were constructed to test the following effects: C_1_, effect of AT (T1 vs. T2), and C_2_, effect of the additive PHY under HS conditions (T2 vs. T3). Respiratory frequency data were analyzed as repeated measures where diet was the between-subject component whereas hour of measurement was the within-subject component. Significant differences were defined when *p* ≤ 0.05, and tendencies were defined when *p* > 0.05 but ≤0.10.

## 3. Results

### 3.1. Ambient and Body Temperature

The average ATs inside the TN and the HS rooms are shown in Figure 1. Inside the TN room, the AT recorded during adaptation and the 8-day experimental periods ranged from 22.6 to 25.2 °C, but it fluctuated from 29.8 to 35.1 °C inside the HS room during the experimental period. The temperature–humidity index inside the TN room was 77 ± 1.9, but it reached 111 ± 6.2 inside the HS room. The BT of pigs housed inside the TN room remained relatively constant, around 39.5 °C, with the exception of the time right after pigs were fed (0700 and 1900 h), when it increased up to 0.5 °C (Figure 2). In contrast, the BT of pigs inside the HS room fluctuated in a similar manner as the AT did, ranging from around 39.6 to 41.2 °C. It increased as the AT also increased, with marked peaks observed right after feeding (0700 and 1900 h). Interestingly, the BT increment observed after the evening meal in the HS pigs fed the phytogenic diet was smaller (0.5 °C) than that of HS pigs fed the control diet (0.8 °C). On average, the BT was higher in HS pigs than in TN pigs (*p* < 0.05). The HS pigs reduced their voluntary physical activity in comparison with the TN pigs, but they remained healthy.

### 3.2. Respiration Rate

The respiration rate, measured as the number of abdomen expansions per min during the morning (0700 h) and afternoon (1700 h), is presented in Figure 3. At 0700 h, the respiration rate did not differ between TN and HS pigs regardless of the diet (*p* > 0.10). However, at 1700 h, the number of abdomen expansions in the HS-C and HS-PHY pigs was higher (*p* < 0.01) than that in TN-C pigs, but it did not differ between the HS-C and HS-PHY pigs (*p* > 0.10). In the TN-C pigs, the morning respiration rate did not differ from the afternoon one.

### 3.3. Performance

The performance results of pigs during the 8-day experimental period are presented in Table 3. The average daily weight gain and feed intake of HS-C pigs decreased in comparison with TN-C pigs (*p* < 0.01). However, the HS-PHY pigs had a higher weight gain (*p* < 0.05) and feed intake (*p* < 0.01) than HS-C pigs. The feed/gain ratio of TN-C pigs was better (*p* < 0.01) than that of HS-C pigs, and that of HS-PHY pigs tended to be better (*p* < 0.10) than that of HS-C pigs.

### 3.4. Antioxidant Activity of Diets and Intestinal Digesta

The free radical scavenging activity (antioxidant activity) of the additive PHY, the control diet without and with PHY, and the ileal digesta of heat-stressed pigs, expressed as a percentage of oxidation inhibition, is presented in Figure 4. The additive PHY had a 56.2% inhibition. The control diet without PHY had 9.2% inhibition, whereas the diet supplemented with PHY had 18.4% inhibition. Antioxidant activity in ileal digesta from HS-PHY pigs (65.6% inhibition) was higher (*p* < 0.01) than that in ilea digesta from HS-C pigs (44.1% inhibition).

### 3.5. Serum Antioxidant Enzyme Activity

The activity of the antioxidant enzymes in serum is presented in Table 4. SOD activity decreased in the HS-C pigs, in comparison with TN-C and HS-PHY pigs (*p* < 0.05). CAT activity was not affected by AT (TN-C vs. HS-C), but it was higher in HS-PHY pigs than in HS-C pigs (*p* < 0.05). GPX was not affected by AT or phytogenic supplementation (*p* > 0.10).

### 3.6. Small Intestine Histomorphology

The histomorphological characteristics of the small intestine epithelium are presented in Table 5. In the duodenum, villus height was not affected by AT or PHY supplementation, but crypt depth was reduced, and the villus height/crypt depth ratio increased in HS-C compared to TN pigs (*p* < 0.01). The jejunum villus height was shorter in HS-C pigs than in both HS-PHY and TN-C pigs (*p* < 0.01). Crypt depth was larger in TN-C pigs than in both groups of HS pigs, but the villus height/crypt depth ratio was greater in HS-PHY pigs compared to HS-C pigs (*p* < 0.01). In the ileum, villus height and crypt depth decreased (*p* < 0.01), but the villus height/crypt depth ratio increased in HS-C pigs, compared to TN-C pigs (*p* < 0.05).

### 3.7. Gene Expression

The relative expression of HSP90 in the liver and the *longissimus dorsi* muscle and the expression of the amino acid transporter CAT-1 in the liver and myosin-II in the longissimus muscle are presented in Figure 5. The HS-C pigs tended to have a higher expression of HSP90 in the liver compared to the TN-C pigs (*p* < 0.10). In muscle, HSP90 expression was higher in the HS-C pigs compared to both the TN-C (*p* < 0.05) and the HS-PHY pigs (*p* < 0.05).

The expression of the amino acid transporter CAT-1 in the liver tended to be higher in HS-C pigs than in TN-C pigs (*p* < 0.10). Compared to HS-C pigs, CAT-1 expression was higher in HS-PHY pigs (*p* < 0.05). Myosin expression in the muscle of HS-C decreased (*p* < 0.05) in comparison with TN-C pigs, but it did not differ among HS-C and HS-PHY pigs (*p* > 0.10).

### 3.8. Serum AA Concentrations

The concentration of some amino acid metabolites in the blood serum of the pigs is presented in Table 6. Serum citrulline, ornithine, GABA, taurine, and urea increased (*p* < 0.01); cystathionine and α-NH2-butyric acid tended to increase (*p* < 0.10); but carnosine decreased (*p* < 0.05) in HS-C pigs in comparison with the TN-C pigs. Adding PHY to the control diet of HS pigs (HS-PHY) decreased serum urea compared to the HS-C pigs.

## 4. Discussion

Several physiological, metabolic, and behavioral alterations that negatively affect performance occur when pigs are exposed to high ambient temperatures (above 30 °C). The greatest impact occurs during the first 7 days of exposure to HS, also defined as the acute phase [3,39]. On the other hand, certain phytogenics, such as capsaicin contained in *Capsicum* spp., apparently have the potential to mitigate the negative impacts of HS on pigs. Thus, the present experiment aimed to determine whether a phytogenic solution based on *Capsicum* spp. could help pigs to counteract some of the negative effects of HS during the acute phase. In the present study, heat-stressed pigs were exposed to ambient temperature exceeding 30 °C and temperature–humidity index (111) well above the index for TN pigs (77) all the time.

Increased BT and respiration rate are two of the first physiological involuntary and behavioral voluntary alterations, respectively, that occur when pigs are suffering from HS [3,40,41]. Indeed, these variables are commonly used to assess whether pigs are exposed to HS conditions. In the present experiment, the BT (around 39.5 °C) and respiratory frequency (around 40 breaths/min) of TN pigs were within the normal range [42]. However, the BT of HS pigs exceeded 40.5 °C, increasing up to 41.4 °C for around 16 h every day, and the respiration rate increased up to 94 breaths per minute during the afternoon. The observed values in the present study confirmed that pigs were subjected to HS conditions most of the time.

Interestingly, the BT of HS pigs fed the phytogenic solution (HS-PHY) was 0.5 °C lower than that of pigs fed the control diet (HS-C) from 19:30 h to around 03:00 h the following day. This BT reduction is noteworthy because (a) it occurred when the daily heat load accumulation was the highest, and (b) that was the time when pigs had the greatest difficulty eliminating body heat. Conserving BT within the normal range, which is under the control of the hypothalamus thermoregulatory system, is essential for maintaining homeostasis and normal functioning of the body [43]. Sensory neurons that innervate the skin and viscera detect changes in both body surface (peripheral) and internal organ temperature due to changes in ambient temperature [44]. According to Morrison and Nakamura [45], warm-sensitive neurons with terminals distributed in the skin become activated when ambient temperature increases through the activation of thermoreceptors located in the plasma membrane and transmit this information to the hypothalamic regulatory system. In turn, the hypothalamus triggers a response oriented towards a reduction in body heat production and/or an increase in body heat dissipation in order to maintain normal BT [44]. Thermoreceptors belong to the transient receptor potential family of ion channels that detect ambient temperature changes, depolarize neurons, and propagate action potentials [46]. These receptors are capable of converting temperature stimuli into electric potentials [47] involved in the transmission of information to the hypothalamus. The transient receptor potential type V member 1 (TRPV1) is the most abundant heat-activated non-selective cation channel in sensory neurons [48]. Importantly, capsaicin is considered to be the prototypical TRPV1 agonist and has pronounced effects on thermoregulation [49,50]. The activation of TRPV1 following capsaicin ingestion has been previously shown to lower body temperature and induce hypothermia [17]. Hence, the reduced BT in the HS pigs fed the diet supplemented with the *Capsicum* spp.-based extract may indicate a hypothalamic feedback response triggered by capsaicin, which promoted a reduction in the BT of pigs exposed to HS conditions.

The production of reactive oxygen species (ROS; e.g., superoxide anion, H_2_O_2_) is an inevitable result of cell metabolism, but the cellular antioxidant system under normal conditions is sufficient to prevent damaging increments in the concentration of ROS [51]. Components of the system, including the antioxidant enzymes CAT, SOD, and GPX, are also used as oxidative stress markers [52]. However, oxidative stress may occur if the equilibrium between antioxidants and ROS is lost in favor of the latter [53], resulting in cell damage [54]. In line with this, it has been reported that HS exposure increases ROS production in avian muscle cells [8] and pigs [41,55]. In the present experiment, in agreement with those reports, the activity of SOD in the serum of the HS-C pigs decreased by about 24% compared to TN-C pigs. Conversely, certain phytogenic compounds, such as those in *Capsicum* spp., possess antioxidant properties [56]. The analyzed antioxidant activity of the phytogenic solution we used in the present experiment was 56% equivalent to that of vitamin E, whereas the antioxidant activity in the control diet without and with the additive was 9.2 and 18.4%, respectively, equivalent to vitamin E. The analyzed antioxidant activity in the digesta of the HS-PHY pigs was 37% higher than that in the digesta of the HS-C pigs, which confirmed that the differential antioxidant activity between the two groups of pigs was conserved once the diet was consumed. Moreover, the serum activity of both superoxide dismutase and catalase was about 25% higher in the HS-PHY than in the HS-C pigs. In fact, the activity of these enzymes did not differ among HS-PHY and TN-C pigs. A similar response was observed in HS pigs fed a diet supplemented with 20% DL-Met above the requirement [41]. In addition to their inherent antioxidant activity, capsaicinoids may enhance the oxidative status in mammalian [57] and avian [58] tissues by activating intestinal TRPV1 receptors and upregulating the expression of uncoupling protein-2, which serves as an adaptive antioxidant defense mechanism [59]. Furthermore, capsaicinoids augment antioxidant activities by activating NF-E2-related factor 2 and enhancing mitochondrial antioxidant enzyme production [58]. These data indicate that the antioxidant activity of phytochemicals contained by *Capsicum* spp., which appeared to be conserved in the digesta, contributed to improving the antioxidant status in the serum of the HS-PHY pigs.

Heat shock proteins (HSPs) protect cells against various stressors, including HS, as they participate in the proper assembly of newly synthesized proteins and protein complexes, the disassembly of protein aggregates, the degradation of misfolded proteins, and the functioning of cellular proteins [60]. HSP90, among all members of the HSP family (HSP100, HSP90, HSP70, HSP60, HSP40, and small ones), plays a crucial role in maintaining normal cellular function under stress conditions [47]. In the present experiment, the abundance of mRNA coding for the synthesis of HSP90 in the longissimus muscle of HS pigs fed the control diet was 58% higher than that of the TN-C pigs. These results are in agreement with other reports from our lab [10,61] and others [3,11,62], showing consistent increments in the expression of HSP because of the exposure of pigs to HS. However, HS pigs fed the diet supplemented with the phytogenic solution had a 56% decrease in the mRNA abundance of HSP90 in muscle, compared to that in HS pigs fed the control diet. Moreover, the mRNA abundance of HSP90 was similar between the TN-C and the HS-PHY pigs. HSP expression is linked to the development of oxidative stress; increased ROS levels lead to higher HSP expression. Thus, the increased antioxidant activity, coupled with the reduced BT, may have contributed to the reduced expression levels of HSP90 in the longissimus muscle of PHY-supplemented pigs [63].

Alterations in the concentration of several metabolites in serum occurred in HS pigs compared to TN pigs, and those changes may be linked to their HS response. For instance, carnosine, a non-enzymatic free radical scavenger and natural antioxidant [64] decreased by 45% in HS-C pigs in comparison with TN-C pigs, reflecting the differences observed in the activities of antioxidant enzymes. Also, GABA is the preferred neurotransmitter used by warm-sensitive neurons [44]; thus, the increased serum concentration of GABA might suggest a higher demand of those neurons for GABA to transmit the information to the hypothalamus. Citrulline and ornithine are precursors for the synthesis of polyamines, whose function consists of stimulating the proliferation of intestinal cells [65], and increased by about 60% in the serum of HS pigs compared to TN pigs. This alteration was also associated with reduced intestinal villus height in HS pigs, suggesting a higher need for both metabolites. Arg-supplemented diets fed to HS pigs provoked a similar response [66]. Serum urea usually results from an imbalance in serum amino acids, with higher serum values indicating excess or deficiencies of one or more indispensable amino acids [67]. Compared to TN-C pigs, serum urea was 2-fold higher in HS-C pigs, suggesting an imbalanced availability of amino acids in HS-C pigs that may be attributed to several factors. Examples include the use of some amino acids such as arginine and methionine as antioxidants [65] or the use of arginine as a precursor for polyamine synthesis [65]. It can be attributed also to the reduced expression of the gene coding for the synthesis of myosin in muscle and the apparently increased amino acid oxidation, as indicated by the higher serum concentration of 3-m-His, a biomarker of muscle proteolysis [68], which coincides with previous reports [61]. However, serum urea in the HS-PHY pigs was 77% lower than that in HS-C pigs. Similarly, myosin expression was not significantly downregulated in the HS-PHY group. These findings could indicate a more efficient utilization of dietary amino acids by the HS-PHY compared to the HS-C pigs for lean tissue accretion.

The exposure of pigs to HS increases blood flow to the periphery (skin) in an attempt to increase body heat loss, but blood flow to internal organs (small intestine) decreases at the same time [6]. This translates into lower nutrient and oxygen supply to those organs that, in combination with the depressed antioxidant status, might damage the intestinal epithelium. Increments in BT of up to 1.5 °C in pigs have been associated with small intestine damage, as evidenced by a significant reduction in intestinal villus height [55,66,69]. In the present experiment, villus height in both the jejunum and ileum of the HS-C pigs was about 10% shorter, and intestinal temperature was 1.8 °C higher than that of the TN-C pigs. However, villus height in the jejunum of HS-PHY pigs was 16% higher than in the jejunum of HS-C pigs. Dietary intake of capsaicin has been previously shown to increase the absorptive surface area of the small intestine, especially in the jejunum of rats [70] and broiler chickens [71], as well as in the ileum of weaned piglets [57]. These effects may be attributed to the antioxidant and anti-inflammatory effects of capsaicin in the intestinal tract [59]. The jejunum is the largest intestinal segment where most of the digestion and absorption of nutrients occurs. Hence, these data indicate adding the phytogenic solution to the diet may help pigs to restore the jejunum villus height observed in the TN-C pigs.

Pigs exposed to HS reduce their voluntary feed intake by 20 to 40% compared to TN pigs [55,72] as part of a hypothalamic response, among others, to reduce body heat production and prevent further increments in BT [44]. In agreement, in the present experiment, the voluntary feed intake of HS pigs fed the control diet was reduced by 40% in comparison with TN-C pigs. However, supplementing the diet with the phytogenic solution helped HS pigs to recover partially their feed intake; the HS-PHY consumed 18% more feed than the HS-C pigs. Although the apparent phytogenic-related decrease in BT of HS-PHY pigs, compared to the HS-C pigs, might explain the higher feed intake of the former pigs, it is still much below the feed intake of TN pigs. In line with the reduction in feed intake, the body weight gain substantially decreased in HS-C pigs, compared to the TN-C pigs; thus, reduced feed intake appears to be the main factor explaining the decreased weight gain. In this regard, it is worth noting that while feed intake was reduced by about 40% in the HS-C pigs compared to the TN-C pigs, body weight gain decreased by about 60%. This differential response between feed intake and weight gain suggests that the reduced weight gain of HS pigs is attributed not solely to the reduced feed intake but also to the other factors. Examples include (a) an apparent deviation of nutrients from growth to a metabolic response that helped pigs fight HS and (b) a growth-related depression of gene expression, as evidenced by the 70% reduction in the expression of the gene coding for myosin in the longissimus muscle. The substantial depression in feed efficiency of HS pigs in comparison with TN pigs seems to support both hypotheses. Similar discrepancies among performance variables were reported previously with pair-fed [11] and ad libitum fed pigs [39].

The weight gain of HS-PHY pigs was 24% higher than that of HS-C pigs, indicating the phytogenic solution’s beneficial effect, mostly on feed intake, as it was 18% higher in the HS-PHY pigs. However, the weight gain of the HS-PHY pigs was still about 45% lower than that of the TN-C pigs. Therefore, these data indicate that several factors other than a lower FI may explain the depressed performance of HS pigs fed diets supplemented or not with the phytogenic solution. It is important to note that the control diet was formulated to meet 100% of the requirement of the first five limiting amino acids (lysine, threonine, methionine, tryptophan, and valine) [32]. However, because HS pigs consumed 30 to 40% less feed than the TN pigs, these amino acids might have become limiting in HS pigs. Actually, we observed an increment in the weight gain of HS pigs fed a diet containing around 25% more of these amino acids or 20% more methionine [41], in comparison with the non-supplemented pigs. Hence, the combination of strategies aimed at counteracting the negative impact of HS on pigs, such as enriching the diet with limiting amino acids combined with the supplementation of specific phytogenic solutions, may be more effective than a single one.

## 5. Conclusions

In conclusion, the results of the present experiment confirm that pigs exposed to HS experience alterations in antioxidant status, intestinal integrity, and performance. These data also indicate that supplementing the diet with a *Capsicum* spp.-based phytogenic solution can help pigs partially counteract the negative impact of short-term exposure to HS on their BT, serum antioxidant status, intestinal integrity, and performance. However, because selecting six pigs per treatment for slaughter with weights closest to the treatment means may not fully represent the within-treatment variation (n = 14), these results should be taken with caution.

## Figures and Tables

**Figure 1 animals-14-00973-f001:**
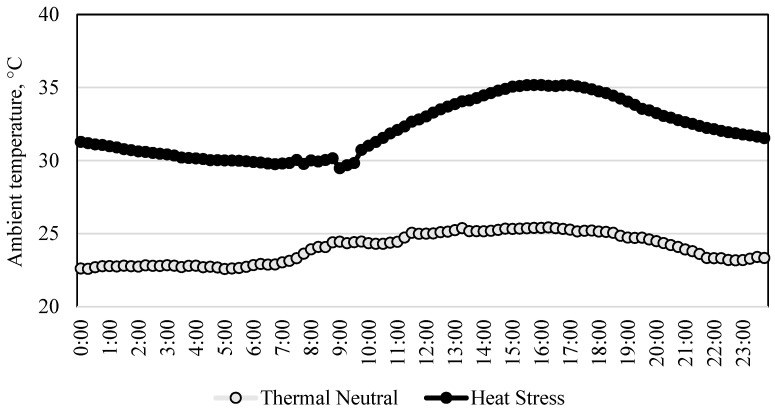
Average ambient temperature inside the thermoneutral and the heat stress rooms during the experimental period.

**Figure 2 animals-14-00973-f002:**
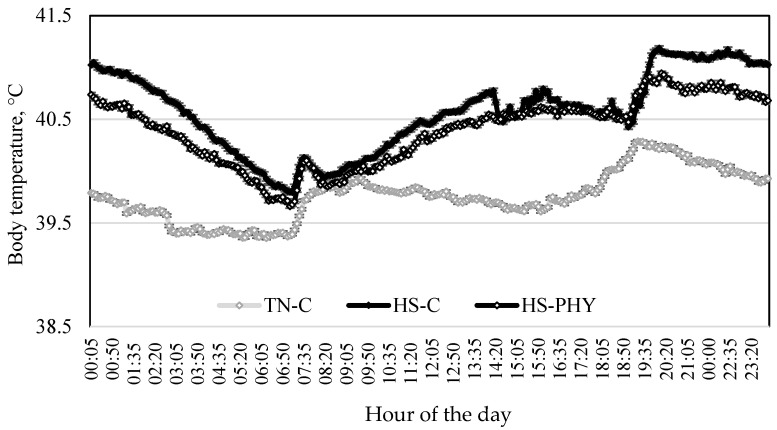
Body temperature of thermoneutral pigs fed a control diet (TN-C) and heat stress pigs fed either the control (HS-C) or the phytogenic-solution-added diet (HS-PHY), average of temperatures along the day during 8-day experimental period.

**Figure 3 animals-14-00973-f003:**
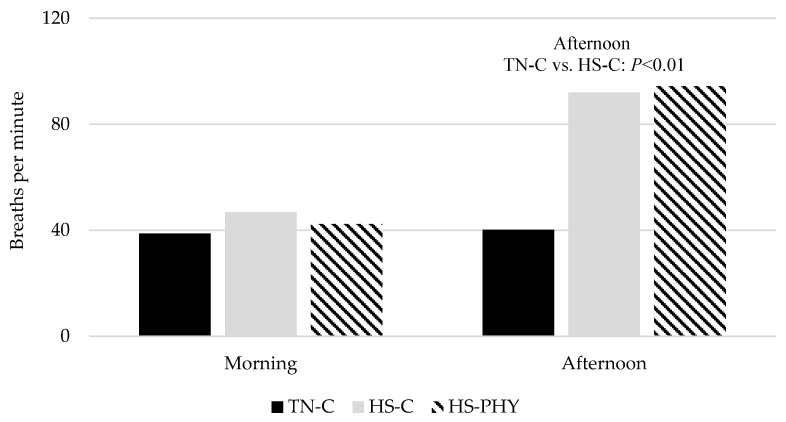
Respiration rate of thermoneutral pigs fed a control diet (TN-C) or heat stress pigs fed either the control (HS-C) or the phytogenic-solution-added diet (HS-PHY), measured during the morning (0700 h) and afternoon (1700) hours.

**Figure 4 animals-14-00973-f004:**
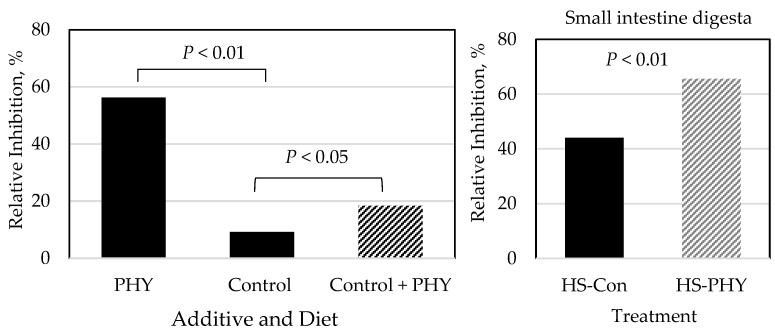
Antioxidant activity in the additive PHY and the control diet with or without PHY (left panel) as well in the ileal digesta of heat-stressed pigs (right panel), expressed as relative inhibition of oxidation using the tocorfosolan (D-Alpha-Tocopheryl Polyethylene Glycol Succinate) as standard.

**Figure 5 animals-14-00973-f005:**
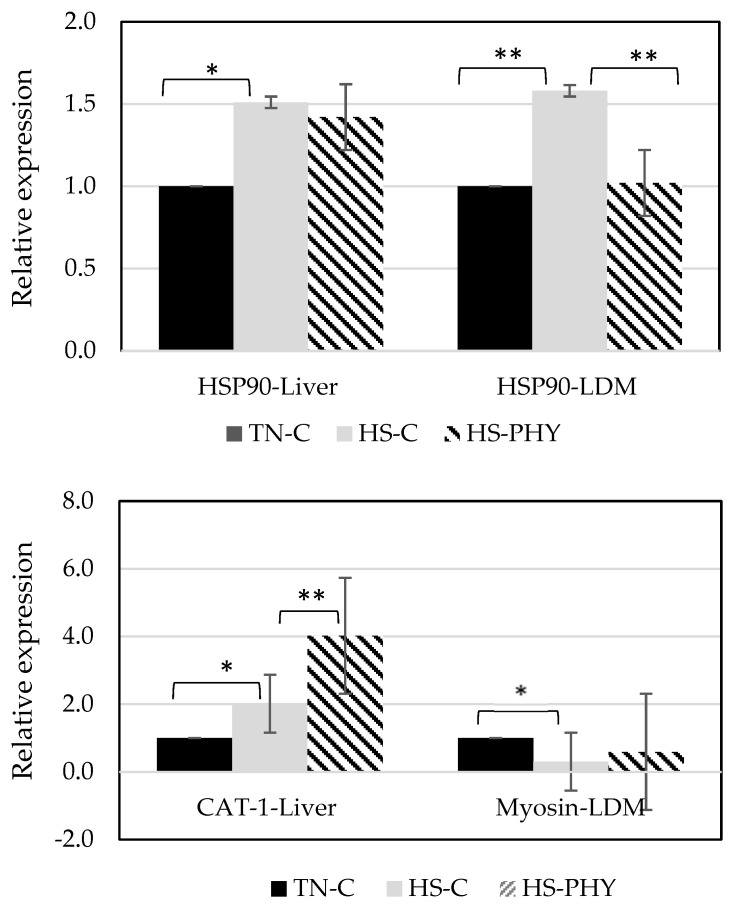
Relative expression of heat shock protein-90 (HSP90) in the liver and the longissimus dorsi muscle (LDM; upper panel) and cationic amino acid transporter (CAT-1) in the liver and myosin in the LDM (lower panel) of pigs exposed to thermoneutral (TNC-C) or heat stress conditions fed either a control diet (HS-C) or this diet supplemented with a phytogenic solution (HS-PHY). * *p* < 0.10; ** *p* < 0.05.

**Table 1 animals-14-00973-t001:** Ingredient composition of the semi-purified experimental diets (%, as-fed basis).

Ingredient	Basal	Additive PHY
Wheat	84.46	84.26
SBM	12	12
L-Lys.HCl	0.54	0.54
L-Thr	0.14	0.14
DL-Met	0.06	0.06
Phytogenic solution ^1^		0.20
Limestone	1.40	1.40
Dicalcium phosphate	0.65	0.66
Iodized salt	0.35	0.35
Vitamin and mineral premix ^2^	0.40	0.40
Calculated content, %		
Crude protein	12.8	12.8
SID Lys	0.99	0.99
SID Thr	0.62	0.62
SID Met	0.28	0.28
Net energy, MJ/kg	10.1	10.1

^1^ Containing *Capsicum* spp. oleoresin with capsaicin as the principal active (ThermoControl^®^). ^2^ Supplied per kg of diet: vitamin A, 4800 IU; vitamin D3, 800 IU; vitamin E, 4.8 IU; vitamin K3, 1.6 mg; riboflavin, 4 mg; D-pantothenic acid, 7.2 mg; niacin, 16 mg; vitamin B12, 12.8 mg; Zn, 64 mg; Fe, 64 mg; Cu, 4 mg; Mn, 4 mg; I, 0.36 mg; Se, 0.13 mg.

**Table 2 animals-14-00973-t002:** Primers used for the quantitative PCR analyses of messenger RNA derived from HSP90, CAT-1, myosin, and RPL4 from pigs.

mRNA	Primer Sequence	Amplicon (bp)
Sus scrofa 90-kDa heat shock protein (HSP90, GenBank: NM_213973.1)		
	Fw 5′GATCACTTGGCTGTGAAGCA3′	470
	Rv 5′TTGAGGGAAACCATCTCGTC3′	
Sus scrofa solute carrier family 7 member 1 (CAT-1, GenBank: NM_001012613.1)		
	Fw 5′CAGATGCTGCGGGTTTGTAA3′	344
	Rv 5′GCGTCACTCATTCTTGCCAT3′	
Sus scrofa myosin, heavy chain 4, skeletal muscle (GenBank: NM_001123141.1)		
	Fw 5′CTTCCCAAGCAGAATCCTTT3′	300
	Rv 5′CTCCTCTCCATCATCTTCC3′	
Sus scrofa ribosomal protein L4 (RPL4, GenBank: DQ845176.1)		
	Fw 5′TGAGCTCTATGGCACTTGGC3′	239
	Rv 5′GAATGGTGTTTCGGCGCATT3′	

**Table 3 animals-14-00973-t003:** Performance of pigs exposed to thermoneutral or heat stress conditions fed either a control diet or this diet supplemented with a phytogenic solution, during both a 10-day adaptation period and the 8-day experimental period.

	Treatment ^1^			Contrast *p*-Value ^2^		
	TN-C	HS-C	HS-PHY	SEM	AT	PHY
Initial body weight, kg	26.4	27.8	27.0	1.0	0.300	0.524
Daily weight gain, kg/d	0.850	0.348	0.457	0.04	0.001	0.048
Daily feed intake, kg/d	1.770	1.057	1.246	0.04	0.001	0.006
Feed/gain ratio	2.082	3.037	2.726	0.42	0.005	0.070

^1^ TN-C, thermoneutral pigs with control diet; HS-C, heat stress pigs with control diet; HS-PHY, heat stress pigs with phytogenic solution added to control diet. ^2^ Contrasts: AT, TN-C vs. HS-C; PHY, HS-C vs. HS-PHY.

**Table 4 animals-14-00973-t004:** Activity (U/mL) of the antioxidant enzymes superoxide dismutase (SOD), catalase (CAT), and glutathione peroxidase (GPX) in the serum of thermoneutral pigs fed a control diet (TN-C) or heat stress pigs fed either the control (HS-C) or the phytogenic-solution-added diet (HS-PHY).

	Treatment ^1^			Contrast *p*-Value ^2^		
	TN-C	HS-C	HS-PHY	SEM	AT	PHY
SOD	2.507	1.907	2.476	1.60	0.018	0.013
CAT	137.1	114.5	152.1	12.2	0.219	0.042
GPX	911	1014	1022	46	0.129	0.904

^1^ TN-C, thermoneutral pigs with control diet; HS-C, heat stress pigs with control diet; HS PHY, heat stress pigs with phytogenic solution added to control diet. ^2^ Contrasts: AT, TN-C vs. HS-C; PHY, HS-C vs. HS-PHY.

**Table 5 animals-14-00973-t005:** Intestinal (duodenum, jejunum, and ileum) morphology of thermoneutral pigs fed a control diet (TN-C) or heat stress pigs fed either the control (HS-C) or the phytogenic-solution-added diet (HS-PHY).

	Treatment ^1^			Contrast *p*-Value ^2^		
	TN-C	HS-C	HS-PHY	SEM	AT	PHY
Duodenum						
Villus height, µm	617	604	617	11.0	0.452	0.424
Crypt depth, µm	362	304	309	1.00	0.001	0.717
Height/depth	1.73	2.07	2.06	0.05	0.001	0.922
Jejunum						
Villus height, µm	644	570	662	12.5	0.001	0.001
Crypt depth, µm	319	275	274	8.6	0.001	0.975
Height/depth	2.09	2.13	2.49	0.06	0.618	0.001
Ileum						
Villus height, µm	554	502	496	12.2	0.005	0.703
Crypt depth, µm	278	224	239	7.4	0.001	0.124
Height/depth	2.04	2.31	2.19	0.07	0.001	0.200

^1^ TN-C, thermoneutral pigs with control diet; HS-C, heat stress pigs with control diet; HS PHY, heat stress pigs with phytogenic solution added to control diet. ^2^ Contrasts: AT, TN-C vs. HS-C; PHY, HS-C vs. HS-PHY.

**Table 6 animals-14-00973-t006:** Concentration of free amino acid metabolites in the serum (µg/mL) of thermoneutral pigs fed a control diet (TN-C) or heat stress pigs fed either the control (HS-C) or the phytogenic-solution-added diet (HS-PHY).

Item ^1^	Treatment			Contrast *p*-Value ^2^		
	TN-C	HS-C	HS-PHY	SEM	AT	PHY
1-m-His	4.0	4.1	3.7	0.5	0.913	0.649
3-m-His	0.8	1.0	1.0	0.1	0.085	0.863
Carnosine	9.6	5.3	7.2	1.4	0.050	0.361
Citrulline	13.9	22.5	21.8	2.1	0.013	0.818
Cystathionine	0.7	2.3	2.3	0.6	0.083	0.989
OH-Lysine	1.3	1.5	1.4	0.2	0.571	0.869
OH-Proline	13.0	11.3	11.1	1.7	0.485	0.917
Ornithine	9.2	14.9	14.7	1.3	0.008	0.902
P-Serine	3.8	3.1	2.7	0.4	0.238	0.493
Sarcosine	2.9	3.5	3.2	0.3	0.166	0.478
AAAA	7.4	11.5	10.4	2.2	0.209	0.724
AABA	1.2	2.4	3.1	0.5	0.100	0.332
β-Alanine	0.6	0.8	1.2	0.3	0.677	0.392
GABA	0.8	1.9	1.5	0.3	0.015	0.310
Taurine	20	26	25	0.9	0.001	0.325
Urea	184	377	289	19	0.001	0.008

^1^ 1-m-His, 1-methyl-histidine; 3-m-His, 3-m-histidine; AAAA, α-NH_2_-adipic acid; AABA, α-NH_2_-butyric acid; GABA, γ-NH_2_-butyric acid. ^2^ Contrasts: AT, TN-C vs. HS-C; PHY, HS-C vs. HS-PHY.

## Data Availability

The data presented in this study are available on request from the corresponding author.

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
