# Peer review of "A Capsaicin-Based Phytogenic Solution Improves Performance and Thermal Tolerance of Heat-Stressed Growing Pigs"

_animals, 2024, doi:10.3390/ani14060973_

Round 1
Reviewer 1 Report
Comments and Suggestions for Authors
Methods should provide more information regarding pig housing, such as floor type, ventilation type and rate, etc. Was the study approved by an animal care committee?
Elements of the design may lead to inflation of significance of results. Three contrasts were tested while there were only 2 degrees of freedom for treatment. The selection for slaughter of 6 pigs per treatment with body weights closest to the treatment mean can be expected to lead to an under-estimation of variation and standard errors.
Interpretation of the results can be affected by the sudden change in ambient temperature for the heat stressed pigs and the short 8-day duration of the study.

Comments on the Quality of English LanguageSentence structure and word choice should be improved for clarity. Some examples are highlighted in the text.
Author Response
L115. Need to describe housing conditions, such as floor type, ventilation and rate, lighting regime.
R. Thanks for your comment, this information is included in the revised version of the manuscript.
L118. Was the study approved by the appropriate animal care and use committee?
R. Thanks for your question. Yes, the approval document is included in the submission package, and ethics information is now included at the beginning of Materials and Methods.
L134. 8 days suggests that the study reflects a short term response to the sudden increase in AT. The effect of a sudden increase in AT may not be the same as a gradual increase in temperature
R. Thanks for your comment. Agree. We and others have observed that the most critical stage of HS is during the first 7 days of exposure to high AT. That is the reason we decided to evaluate the effect of the herbal extract during this short term.
L143. Feed intake; ad libitum or restricted?
R. Thanks for the comment. The information may be confusing. We wanted to emphasize that fresh feed was offered every day and that the amount of feed supplied daily was in excess, which means that there was always feed in the feeder. The text was modified for better clarity
L147 and 229. Leading to an underestimate of standard error.
R. Thanks for the comment. The rationale for using three contrasts was as follows: We could have had only two contrast, by which we would have compared the effect of HS exposure (C1, TN-C vs. HS-C) and the effect of PHY supplementation (C2, HS-C vs. HS-PHY). However, this set of contrasts does not allow us to know whether any eventual performance or physiological response of HS pigs to supplemental PHY matches any of those in TN pigs. Thus, we planned to have a third contrast that allowed us to know if the response of HS pigs to the PHY helped them to completely recover the performance and physiological variables, compared to the TN pigs.
There was also the option of performing multiple comparison analysis using Tukey, but this procedure only tells us whether the difference between two means are or are not significant at the P<0.05 or whether there is or there is not a tendency at the P<0.10. By the contrary, the use of contrasts produces the actual P-value, which can be more useful for the interpretation of the results.
With regards to the standard error, based on your comment we run the comparison analysis including only 2 contrasts. The standard error of the contrast was exactly the same as when the analysis included 3 contrasts.
L274. Decreased instead of reduced.
R. Thanks. Done
Table 3. Duration of the adaptation period.
R. Thanks. Done
L274. Decreased instead of reduced.
R. Thanks. Done
Table 3. Include units.
R. Thanks. Done
Table 5. Include units.
R. Thanks. Done
Reviewer 2 Report
Comments and Suggestions for Authors
This manuscript presents a comprehensive study on the effects of dietary supplementation with a Capsicum spp.-based phytogenic solution on pigs under heat stress conditions. The study explores an innovative dietary intervention to alleviate heat stress in pigs, a significant concern in swine production. It provides a thorough investigation of various physiological and metabolic parameters, including body temperature, feed intake, antioxidant enzyme activity, and intestinal morphology. The experimental design, including the control and treatment groups, is well-structured, and the methods for data collection and analysis are rigorously defined. The study presents significant findings on the positive impact of the phytogenic solution on improving pigs' thermal tolerance, antioxidant defenses, and performance under heat stress conditions.
However, the study focuses on short-term outcomes, with less emphasis on the long-term effects of capsaicin supplementation on pigs' health and performance. While the data analysis is comprehensive, there might be a need for a more critical discussion on the potential biases or limitations in interpreting the results.
The manuscript could benefit from a comparison with other heat stress mitigation strategies to contextualize the effectiveness of the capsaicin-based solution.
There is limited discussion on the environmental implications of using phytogenic solutions and the ethical considerations regarding the welfare of heat-stressed animals.
Overall, the manuscript contributes valuable insights into dietary interventions for heat stress in pigs, with clear implications for animal welfare and production efficiency. However, addressing the mentioned weaknesses could strengthen the impact and applicability of the research findings.
Line 124-128: Please provide the geographical location of the experimental pig house, including latitude and longitude. Because different climatic environments have different effects on heat stress.
Line 129 and line 143: “All pigs had free access to feed”, but in line 143, “The feed was offered twice a day, at 0700 and 1900 h, in equal amounts time, but with approximately 30% in excess of their predicted daily FI.” Free feeding means that pigs can feed at any time, while feeding twice a day means that there is no feed to eat for a period of time in between. This is inconsistent.
Line 139-140: “The additive PHY consisted of a standardised plant phytogenic solution containing Capsicum spp. Oleoresin with capsaicin being the principal active PSM (trade name ThermoControl® )” The content information of the main components and key active components of the added PHY is missing.
Table 1: The content information of the main components and key active components of the added Phytogenic solution is missing. If it is a commercialized product (trade name ThermoControl®), information such as product name, batch number and manufacturer should be provided to make this trial reproducible.
Figure 1: Besides the temperature, we also need to consider the influence of humidity. Because temperature and humidity work together on the thermal stress response of animals. It is suggested that this graph should be change to temperature and humidity index (THI). This index is widely used to evaluate the heat stress of animals.
Table 3: Initial body weight in line 122 “around 25.0 kg”, but here “36 to 38 kg”. Inconsistent. How old were these pigs? How many days was the trial period?
Author Response
This manuscript presents a comprehensive study on the effects of dietary supplementation with a Capsicum spp.-based phytogenic solution on pigs under heat stress conditions. The study explores an innovative dietary intervention to alleviate heat stress in pigs, a significant concern in swine production. It provides a thorough investigation of various physiological and metabolic parameters, including body temperature, feed intake, antioxidant enzyme activity, and intestinal morphology. The experimental design, including the control and treatment groups, is well-structured, and the methods for data collection and analysis are rigorously defined. The study presents significant findings on the positive impact of the phytogenic solution on improving pigs' thermal tolerance, antioxidant defenses, and performance under heat stress conditions.
The authors thank you very much for the effort you put into this review. All of your comments, suggestions and questions were taken into consideration, which helped to improve the quality of the manuscript.
However, the study focuses on short-term outcomes, with less emphasis on the long-term effects of capsaicin supplementation on pigs' health and performance. While the data analysis is comprehensive, there might be a need for a more critical discussion on the potential biases or limitations in interpreting the results.
R. Thanks for this interesting comment. Attended. It has been reported that the acute phase is the most critical stage for pigs exposed to heat stress (e.g., Pearce et al., 2013; 2014; Yu et al., 2010; Morales et al., 2016). Pigs tend to adapt to heat stress conditions after being exposed to high ambient temperature, and have a partial recovery compared to what is observed under thermoneutral conditions.
The manuscript could benefit from a comparison with other heat stress mitigation strategies to contextualize the effectiveness of the capsaicin-based solution.
R. Thanks for this comment. Done.
There is limited discussion on the environmental implications of using phytogenic solutions and the ethical considerations regarding the welfare of heat-stressed animals.
R. Thanks for the comment. There is no available information regarding the environmental implications of using the phytogenic solutions but, because its inclusion in the diet is very low, we do not expect any significant implication. Concerning the welfare of heat-stressed animals, we mentioned in both the Introduction and Discussion sections the way heat stress affects the welfare of pigs.
Overall, the manuscript contributes valuable insights into dietary interventions for heat stress in pigs, with clear implications for animal welfare and production efficiency. However, addressing the mentioned weaknesses could strengthen the impact and applicability of the research findings.
R. Thanks, this was included in the discussion.
Line 124-128: Please provide the geographical location of the experimental pig house, including latitude and longitude. Because different climatic environments have different effects on heat stress.
R. Thanks for the suggestion, geographical location is included in the new version of the manuscript.
Line 129 and line 143: “All pigs had free access to feed”, but in line 143, “The feed was offered twice a day, at 0700 and 1900 h, in equal amounts time, but with approximately 30% in excess of their predicted daily FI.” Free feeding means that pigs can feed at any time, while feeding twice a day means that there is no feed to eat for a period of time in between. This is inconsistent.
R. Thanks for the comment. It may be confusing but it is not inconsistent; we apologize for partially disagreeing with you. We wanted to emphasize that fresh feed was offered every day and that the amount of feed supplied daily was in excess, which means that there was always feed in the feeder. The text was modified for better clarity.
Line 139-140: “The additive PHY consisted of a standardised plant phytogenic solution containing Capsicum spp. Oleoresin with capsaicin being the principal active PSM (trade name ThermoControl® )” The content information of the main components and key active components of the added PHY is missing.
R. Thanks for this comment. The producer did not provide this information as this is a patented product.
Table 1: The content information of the main components and key active components of the added Phytogenic solution is missing. If it is a commercialized product (trade name ThermoControl®), information such as product name, batch number and manufacturer should be provided to make this trial reproducible.
R. Thanks for this comment. The product name is ThermoControl and the producer is DELTAVIT - CCPA GROUPE, France.
Figure 1: Besides the temperature, we also need to consider the influence of humidity. Because temperature and humidity work together on the thermal stress response of animals. It is suggested that this graph should be change to temperature and humidity index (THI). This index is widely used to evaluate the heat stress of animals.
R. Thanks for the suggestion. Agree. Values of the THI are included in the revised manuscript. Although relative humidity contributes to provoke HS, ambient temperature is the main factor causing it. Thus, we prefer to keep the ambient temperature graph as well including the THI values.
Table 3: Initial body weight in line 122 “around 25.0 kg”, but here “36 to 38 kg”. Inconsistent. How old were these pigs? How many days was the trial period?
R. Thanks for the comment and questions. Agree. There are some errors in the Materials and Methods section as well as in Table 3. We apologize for that. The revised version has the correct values. The adaptation period was 10-d long, and the trial period was 8-d long.
Reviewer 3 Report
Comments and Suggestions for Authors
The title of the work correctly reflects the meaning of the experiment.
The summaries briefly and concisely describe the research conducted and present the obtained results synthetically.
Introduction
The introduction presents the most important aspects related to the subject of the manuscript in a precise and synthetic way. A wide variety of literature sources were used to support the information provided. The purpose and research hypothesis clearly and transparently reflect the meaning of the planned experience.
L52-54: where does the knowledge and assumption come from that most pig production takes place in regions where they are exposed to heat stress? Who is behind this information? And who is this Anonymous (2015)?
Material and methods
L112: In what country and city was the experiment conducted? Please provide this, because the authors come from three countries and two different continents and this is what is missing here.
L125: is 22±2°C a thermoneutral temperature for pigs weighing 25 kg? In the country I come from, the optimal temperature for such animals is 19 ° C. Please explain why such a choice?
Apart from the above-mentioned comments, the remaining research methods were correctly described and characterized.
Results
Table 3 shows the results regarding production results. The body weight at the beginning of the experiment is approximately 37 kg, and in L122 it was recorded as 25 kg? Why this difference? Can you somehow explain this?
What explains such a significant decrease in body weight gain in pigs in the experimental groups? Did heat stress itself have such a significant impact on these results?
The layout of tables and charts is clear and transparent. I have no objections to the presented research results.
Discussion
I have no objections to the discussion presented in the manuscript. The results of own research were properly discussed with the available literature on the topic.
Summary
The summary briefly and synthetically summarizes the results obtained and indicates the need to use phytogenic additives for animals living in heat stress.
Author Response
The title of the work correctly reflects the meaning of the experiment.
The summaries briefly and concisely describe the research conducted and present the obtained results synthetically.
R. Thanks for your pleasant comments.
Introduction
The introduction presents the most important aspects related to the subject of the manuscript in a precise and synthetic way. A wide variety of literature sources were used to support the information provided. The purpose and research hypothesis clearly and transparently reflect the meaning of the planned experience.
R. Thanks for your pleasant comments.
L52-54: where does the knowledge and assumption come from that most pig production takes place in regions where they are exposed to heat stress? Who is behind this information? And who is this Anonymous (2015)?
R. Thanks for the questions; we included a reference for this issue at the new version of the manuscript.
Material and methods
L112: In what country and city was the experiment conducted? Please provide this, because the authors come from three countries and two different continents and this is what is missing here.
R. Thanks for the question. Done; the geographical location was included in the revised manuscript.
L125: is 22±2°C a thermoneutral temperature for pigs weighing 25 kg? In the country I come from, the optimal temperature for such animals is 19 ° C. Please explain why such a choice?
Apart from the above-mentioned comments, the remaining research methods were correctly described and characterized.
R. Thanks for the question. According to the Federation of Animal Science Societies (2010), the TN zone of growing pigs ranges from 15 to 25 °C, being the lowest temperature for finishing pigs (>70 kg) and the highest one for young pigs (<15 kg). Several authors (e.g., Collin et al., 2001; Renaudeau et al., 2010) have shown that the thermoneutral temperature for pigs within the 25 to 50 kg range is around 22 C. This is why we chose this value as the thermoneutral ambient temperature. Reference was included in the document.
Results
Table 3 shows the results regarding production results. The body weight at the beginning of the experiment is approximately 37 kg, and in L122 it was recorded as 25 kg? Why this difference? Can you somehow explain this?
R. Thanks for pointing out the difference and for the questions. Agree, those values are wrong. In L122, it should say: “…the average initial body weight was around 27 kg”. The initial body weights shown in Table 3 are also incorrect. Both errors were corrected in the revised manuscript.
What explains such a significant decrease in body weight gain in pigs in the experimental groups? Did heat stress itself have such a significant impact on these results? The layout of tables and charts is clear and transparent. I have no objections to the presented research results.
R. Thanks for the question. Several factors explain the marked reduction in body weight gain of pigs exposed to heat stress. First, there is an increase in body temperature as shown in Figure 2. Then, as a means to prevent further increments in body temperature, heat stressed pigs reduced their voluntary feed intake, up to 50%. This is the main reason why pigs gain less weight. In addition, heat stressed pigs utilize some nutrients such as amino acids to fight its negative effects, thus deviating those nutrients from growth. This type of decrease in body weight gain is very consistent and has been reported by several authors.
Discussion
I have no objections to the discussion presented in the manuscript. The results of own research were properly discussed with the available literature on the topic.
R. Thanks for your comments.
Summary
The summary briefly and synthetically summarizes the results obtained and indicates the need to use phytogenic additives for animals living in heat stress.
R. Thanks for your comments.
Round 2
Reviewer 1 Report
Comments and Suggestions for Authors
To clarify my comment regarding the number of contrasts: increasing the number of contrasts over the number of degrees of freedom for error does not change the standard error, rather it increases the probability of finding a significant difference when there is none. The formula is 1-((1-0.05)^3/2) = 0.074 where 0.05 is the desired P, 3 is the number of contrasts and 2 is the degrees of freedom for treatment and 0.074 is the actual P value. This leads to ever-estimation of significant differences. This can be fixed by using P = 0.074 as the test for significance.
My comment regarding under-estimation of standard errors refers to selecting pigs for slaughter that were closest to the treatment means. These pigs do not represent the full range of variability within treatment, resulting in under-estimation of standard errors for all measurements taken on these pigs and spurious findings of significance. I do not have a mathematical fix for this. I can only suggest a statement noting the the problem and recommending that results must be interpreted with caution.
Comments on the Quality of English LanguageWhile I included some notes on language, these were not exhaustive.
Author Response
To clarify my comment regarding the number of contrasts: increasing the number of contrasts over the number of degrees of freedom for error does not change the standard error, rather it increases the probability of finding a significant difference when there is none. The formula is 1-((1-0.05)^3/2) = 0.074 where 0.05 is the desired P, 3 is the number of contrasts and 2 is the degrees of freedom for treatment and 0.074 is the actual P value. This leads to ever-estimation of significant differences. This can be fixed by using P = 0.074 as the test for significance.
R. Thank you very much for the clarification. Thanks also for suggesting a way to fix the over-estimation. Because contrasting TN-C vs. HS-C and HS-C vs. HS-PHY are the most meaningful comparisons in the present study, we rather decided to keep these two contrasts only, and to eliminate the contrast TN-C vs. HS-PHY. The whole manuscript was revised accordingly.
My comment regarding under-estimation of standard errors refers to selecting pigs for slaughter that were closest to the treatment means. These pigs do not represent the full range of variability within treatment, resulting in under-estimation of standard errors for all measurements taken on these pigs and spurious findings of significance. I do not have a mathematical fix for this. I can only suggest a statement noting the problem and recommending that results must be interpreted with caution.
R. Thank you for the suggestion. A statement was included in the conclusion of the revised manuscript.
Comments on the Quality of English Language. While I included some notes on language, these were not exhaustive.
R. Thank you. Done